# Fabrication, Characterization and Drainage Capacity of Single-Channel Porous Alumina Ceramic Membrane Tube

**DOI:** 10.3390/membranes12040390

**Published:** 2022-03-31

**Authors:** Jianzhou Du, Xin Xiao, Duomei Ai, Jingjin Liu, Long Qiu, Yuansheng Chen, Kongjun Zhu, Luming Wang

**Affiliations:** 1School of Materials Science and Engineering, Yancheng Institute of Technology, Yancheng 224051, China; dujianzhou123@163.com (J.D.); xiaoxin0627@163.com (X.X.); 15052323525@163.com (D.A.); qiulong233@163.com (L.Q.); chenys@ycit.edu.cn (Y.C.); 2State Key Laboratory of Mechanics and Control of Mechanical Structures, Nanjing University of Aeronautics and Astronautics, Nanjing 210016, China; 3School of General Education, Wuchang University of Technology, Wuhan 430223, China; liuliza@163.com

**Keywords:** porous Al_2_O_3_ ceramic membrane tube, extrusion molding process, sintering temperature, grain size

## Abstract

The single-channel Al_2_O_3_-based porous ceramic membrane tubes (PCMT) were prepared with different grain size of Al_2_O_3_ powders by extrusion molding process, combing the traditional solid-phase sintering method. The effects of raw grain size and sintering temperature on the microstructure, phase structure, density, and porosity were investigated. The results revealed that with further increase in sintering temperature, the density of porous ceramics increases, while the porosity decreases, and the pore size decreases slightly. The pore size and porosity of porous ceramics increase with the increase in raw grain size, while the density decreases. Future, in order to study the water filtration of PCMT, the effect of porosity on the pressure distribution and flow velocity different cross-sectional areas with constant feed mass flow was analyzed using Fluent 19.0. It was found that an increase in the porosity from 30% to 45% with constant feed mass flow influenced transmembrane pressure, that varied from 216.06 kPa to 42.28 kPa, while the velocity change at the outlet was not obvious. Besides, it was observed that the surface pressure is almost constant along the radial direction of the pipe, and the velocity of water in the PCMT is increasing with the decreasing of distance to the outlet. It was also verified that the porosity being 39.64%, caused transmembrane pressure reaching to 77.83 kPa and maximum velocity of 2.301 m/s. These simulation and experimental results showed that the PCMT have good potential for water filtration.

## 1. Introduction

Due to excellent stability, mechanical strength and service life, the ceramic membranes have been widely applied in petroleum and chemical industry, medical treatment, water treatment and other fields. The rapid development of membrane technology leads the increasing of membrane in water purification and wastewater treatment industry. Porous ceramic membranes possess combined advantages of better chemical stability and mechanical strength. It can be applied in harsh operating conditions such as high temperature and the presence of corrosive chemicals [1,2,3,4,5].

In the water purification application, the ceramic membrane separation technology is more effective than traditional methods, such as activated carbon adsorption, photocatalytic treatment and so on [6,7]. At present, a variety of preparation processes, such as extrusion molding process [8,9], dip-coating process [10,11] and gel-casting process [12,13] are employed to produce PCMT. The main types of membranes include polymeric and ceramic membranes owing to the material of construction [14]. Although ceramic membranes are generally more expensive than polymeric membranes, they have the advantage in mechanical strength and durability. The ceramic membranes are usually made of inorganic materials such as aluminum oxides [15,16], silicon carbide [17,18] and so on. Zou et al. [11] designed high performance double-layer α-alumina ultrafiltration membrane using co-sintering process. The pure water permeability of membranes exhibited 70 Lm^−2^h^−1^bar^−1^. Rashad et al. [19] prepared porous ceramic membranes with good water flux, high porosity, and flexural strength at 1400 °C by using clay and aluminum fluoride trihydrate as row materials and adding 10 wt.% alumina. In Ref. [20], porous alumina ceramic membrane with maximum porosity of 57% was prepared by sol-gel method using corn starch as pore forming agent. Qin et al. [21] prepared γ-Al_2_O_3_ ultrafiltration membranes using Sol-gel process, indicating that the grain size, distribution, and morphology affected the pore size, pore size distribution, porosity and pore volume of the membranes. In Ref. [22], a low-cost alumina-mullite composite hollow fiber ceramic membrane was fabricated via phase-inversion method followed by high temperature sintering. The porosity of its surface porosity achieved 5% at a sintering temperature of 1300 °C. Wang et al. [23] prepared porous-foam mullite-bonded SiC-ceramic membranes as high temperature filter by a direct foaming method combined with gel-tape casting. The porosity and bending strength were 69.2~84.1% and 4 MPa, respectively. Liu et al. [24] studied the effect of SiC grain size on the permeability of porous SiC ceramics. It was found that the permeability increased from 6.65 × 10^−15^ m^2^ to 4.85 × 10^−13^ m^2^ as average grain size increased from 4.85 μm to 86.09 μm. In Ref. [25], the influence of different temperature on porous SiC ceramic membrane was studied. With the sintering temperature ranging from 600 °C to 1000 °C, the average pore size increased from 0.58 μm to 0.75 μm, and the porosity decreased from 47.4% to 39.4%. Huang et al. [26] prepared highly permeable porous SiC ceramic membrane by one-step freeze-casting process. Increasing the coarse particle content can improve the permeability efficiency, when the content was 90%, the maximum water permeability reached up to 5.67 × 10^5^ L/m^2^hbar.

Computational fluid dynamics (CFD) technology has been rapidly developed in the simulation of membrane separation and reaction processes. Guilherme et al. [27] studied the effect of process parameters of ceramic membrane modules on oil-water separation using CFD and concluded that increasing feed mass flow could increase transmembrane pressure but increasing membrane permeability would reduce transmembrane pressure. Tang et al. [28] established a complex model that characterizes the tortuosity and nonuniformity of the membrane channel with pores and throats. In Ref. [29], a novel CFD-based method was developed for predicting pressure drop and dust cake distribution of ceramic filter during filtration process. Li et al. [30] simulated a microporous tubular microchannel reactor (MTMCR) using computational fluid dynamics (CFD). It was concluded that the interfacial area and pressure drop increased with the increase in liquid flow rate and with the decrease in micropores size and annular channel width. Yue et al. [31] calculated and analyzed the filtration efficiency and pressure drop of fiber filters with fiber diameters between 10 μm and 20 μm and solid volume fractions (SVFs) between 12% and 19%.

In this study, the extrusion molding process combined with the traditional solid-phase sintering method were adopted to prepare the single-channel Al_2_O_3_-based PCMT. The effects of different grain size of Al_2_O_3_ powders and sintering temperature on the microstructure and properties of Al_2_O_3_-based PCMT were discussed. In order to study the water filtration behavior, the hydrodynamics of water in PCMT was analyzed with Fluent software. The effect of porosity on the pressure distribution on the cross-section of PCMT and the flow velocity at the outlet with constant feed mass flow was investigated. The pressure, flow field and velocity distributions of Al_2_O_3_-based PCMT with different cross-sectional areas were also discussed.

## 2. Preparation and Characterization

### 2.1. Preparation of Al_2_O_3_ Porous Ceramic Membrane Tube

The proposed single-channel Al_2_O_3_-based PCMT were prepared by extrusion molding process combined with the solid-phase sintering method. The α-Al_2_O_3_ powders with different grain sizes namely, 1500#, 300~325#, 150~250#, 80~150# and kaolin were employed as the starting powders. Carbon powders and SiO_2_-Yb_2_O_3_ are the pore-forming agent and sintering aids. Carboxymethyl cellulose (CMC) and dibutyl phthalate (DBP) are the binder and the dispersing agent. Oleic acid and deionized water are the lubricant and the solvent, respectively.

The schematic diagram of Figure 1 is the preparation process of the single-channel Al_2_O_3_-based PCMT. Firstly, the calculated amount of alumina powders, 25 wt.% of kaolin, 8 wt.% of carbon powders and a certain amount of SiO_2_-Yb_2_O_3_ were intensively mixed by the V-type mixer for 5 h. Then the mixture added with 2 wt.% of DBP, 2 wt.% of CMC, 2 wt.% of oleic acid and 40 wt.% of deionized water were stirred for 3 h in a twin-shaft mixer. The ceramic paste was refined by vacuum, then aged in a sealed container at a temperature of 30 °C and a humidity of 45% for 3 days. The single-channel ceramic tube embryo was formed by a ceramic vacuum extrusion molding machine at a constant pressing speed. After dried for 24 h at room temperature, the dried samples were sealed in an alumina crucible for lead atmosphere protection. They were then sintered at different temperature ranging from 1100 °C to 1200 °C for 3 h. The interval of sintering temperature is 50 °C and its ramping rate is 2 °C/min.

### 2.2. Characterization

The crystal phase structures of sintered samples were investigated by X-ray diffraction (X’ PERT3 POEDER, PANalytical Co., Eindhoven, The Netherlands), and its scanning range 2θ is from 10° to 70°. According to Archimedes principle, the apparent porosity (*P*) and bulk density (*ρ*) of the sintered samples were measured by automatic electronic densitometer (Type ZMD-2, Shanghai Fangrui Instrument Co., Ltd., Shanghai, China) with deionized water as immersion medium. The particles of raw materials were measured individually by the laser granularity analyzer (Type. LS13320, Beckman Coulter, Inc., Brea, CA, USA) with the dry powder system. The precise particle size distributions were developed by the Particle Size Analyzer V6.01. The surface microstructures of raw materials and sintered samples were examined by a scanning electron microscope (SEM, Nova Nano SEM 450, FEI Co., Hillsboro, OR, USA), which was combined with energy-dispersive-X-ray spectroscopy for elemental analysis. The pore size distribution was analyzed by way of Nano Measure software (Ver. 1.2, Fudan University, Shanghai, China).

### 2.3. Modeling Configuration and Grid Meshing

To analyze the performance of PCMT, it was modeled in SolidWorks software and simulated in Fluent software. As the sample of tube shown in Figure 2a, the single-channel tube model was established as shown in Figure 2b. Its length L, inner diameter D, outer diameter and film thickness are 180 mm, 28 mm, 43 mm and 7.5 mm, respectively. To analyze the water filtration, the fluid domain model shown in Figure 2c was developed to simulate the liquid through micropores in porous tube. The green part is the porous medium and the yellow part is the fluid cavity. The outer surface of the tube was set as the fluid inlet. At the ends of sap cavity, one was outlet and another was set as wall. As shown in Figure 2d, the water filtered by porous tube was simulated with Ansys Fluent 19.0 and the number of meshes was 227,088.

## 3. Results and Discussion

### 3.1. Phase Structure and Microstructure

XRD patterns of Al_2_O_3_ powders with grit designation of 1500# are shown in Figure 3a. The major crystalline in the sample was α-Al_2_O_3_ (PDF #99-0036), but no additional crystal phases were detected. Figure 4a shows the XRD patterns of Al_2_O_3_-based PCMT with different grain sizes of alumina powders after sintering process at 1150 °C. The major crystalline phase in all the sintered samples was α-Al_2_O_3_ (PDF #99-0036), containing a little anorthite (CaAl_2_Si_2_O_8_) and spinel (MgAl_2_O_4_) phases. It indicates that the diffraction peak intensity of Al_2_O_3_ becomes higher and the crystallinity becomes better with the decrease in grain sizes of α-Al_2_O_3_ powders. Besides, the characteristics peak position of porous ceramics prepared by alumina powder with different grain sizes is almost unchanged.

Figure 3b,d show the SEM images of Al_2_O_3_ powders with the minimum and maximum grain sizes of the raw materials, respectively. The median particle size of Al_2_O_3_ powders with grit designation of 1500# was 4.191 μm as shown in Figure 3c. Figure 3e,f show the SEM images of raw particles including kaolin particles and carbon powders.

Figure 4b shows the digital photography of Al_2_O_3_-based PCMT. To analyze the changes in pore size distribution, the microstructure of them prepared by alumina powders with different grain sizes sintered at 1200 °C as shown in Figure 4c–f. With the addition of different grain size of alumina powders, pores of different sizes were formed on the surface of ceramic tubes and the pore size is increased with the grain size of alumina powders.

Figure 5a–c show the surface SEM images of Al_2_O_3_-based PCMT sintered at different temperatures. The pores on the surface of PCMT show irregular shape distribution, due to the addition of kaolin. As the sintering temperature increases from 1100 °C to 1200 °C, the average pore diameters range from 5.01 μm to 4.51 μm. Figure 5d shows thermogravimetric and differential analyses of green sample, showing the changes of the weight and the thermal energy with the temperature. The weight loss decrease observed from 250 °C to 450 °C is mainly due to the volatilization of substances by heat, and the green sample is discharged at 550 °C. The PCMT with grit designation of 150~250# sintered at 1200 °C was selected for elemental analysis in Figure 5e. The elemental distributions in Figure 5f–i show that particles were composed of O, Al and Yb.

### 3.2. Porosity and Density of Ceramic Membrane Tubes

Figure 6 illustrates the relationship between density, porosity, sintering temperature, and grain size of PCMT. As shown in Figure 6a, the density increases gradually from 1.4 g·cm^−3^ to 2.2 g·cm^−3^ with the sintering temperature increasing from 1100 °C to 1200 °C and the grain size decreasing from 120 μm to 10 μm. As shown in Figure 6b, the porosity variation was measured in different grain size and sintering temperature. It indicates that the porosity increases with the grain size and decreases with sintering temperature. When the grain size and sintering temperature were 125 μm and 1100 °C, the porosity reached the maximum of 48.4%. Additionally, the minimum porosity was 27.8% with the grain size of 10 μm and sintering temperature at 1200 °C.

### 3.3. Simulation Analysis of Water Filtering

The viscous loss term K_perm_ and an inertial loss term K_loss_ are significant parameters in the simulation of water filtering. In the analysis of gas-liquid flow characteristics of microporous channels in Ref. [30], they were defined as:$$K_{perm}=\frac{D_{p}^{2}}{150}\frac{\varepsilon^{3}}{{\left(1-\varepsilon \right)}^{2}}$$
$$K_{loss}=\frac{3.5}{D_{p}}\frac{\left(1-\varepsilon \right)}{\varepsilon^{3}}\;$$
$$D_{p}=\frac{3d_{h}\left(1-\varepsilon \right)}{2\varepsilon }$$
where *ε* is porosity, *D_p_* is average grain size and *d_h_* is average diameter of micro pores.

In this calculation, the fluid inlet and outlet were selected as the boundary, and all the walls were chosen as standard wall. The feed mass flow rate of inlet was 1 kg/s, and the outlet was under the standard atmospheric pressure. As the fluid media of water, its density is 998.2 kg/m^3^ and viscosity is 1.003 × 10^−3^ Ns/m^2^. At the environment temperature of 300 K, the influence of gravity was ignored. With Pressure-Based solver and Viscous-Realizable k-e model, the pressure and velocity coupling were carried out by SIMPLEC algorithm.

#### 3.3.1. Comparison of Flow Pressure and Velocity Distribution under Different Porosity

The pressure distribution on the cross-section of PCMT and the flow velocity at the outlet were analyzed with the porosity range from 30% to 45%. Figure 7 shows distribution of flow pressure on the cross-section of PCMT, and it can be found that the pressure is symmetrically distributed around the circumference. The Pressure is higher near the inlet of tube, and gradually decrease from the periphery to the center. At porosity of 30%, the maximum pressure is 261.05 kPa and minimum pressure is 44.98 kPa as shown in Figure 7a. As shown in Figure 7d, the maximum pressure is 50.73 kPa and minimum pressure is 8.45 kPa when the porosity is 45%. Obviously, the minimum pressure is at the inner wall of PCMT.

With the different porosity ranging, the flow velocity distribution nephogram at the outlet is shown in Figure 8. The maximum flow velocity is 2.31 m/s at the center of outlet surface, and the velocity decreases from the center to surrounding which minimum flow velocity is 1.24 m/s. The flow velocity does not change with porosity value, that is, the flow velocity is almost not affected by the porosity.

When the flow rate at inlet is constant, the pressure on the cross-section of PCMT decreases with the increasing of porosity, but the flow velocity is hardly affected by the porosity. In addition, transmembrane pressure is significantly influenced by the porosity. When the porosity is from 30% to 45%, the transmembrane pressure decreases from 216.06 kPa to 42.28 kPa.

#### 3.3.2. Pressure, Flow Field, and Velocity Distribution in the Fluid Domain Model

To characterize water filtering, a Al_2_O_3_-based PCMT with a grit designation of 150~250# was selected and sintered at the temperature of 1200 °C. The simulation of pressure, flow field and velocity distributions of porous alumina ceramic membrane tubes with different cross-sectional areas are illustrated in Figure 9.

In the simulation, the porosity is 39.64%, and feed mass flow rate of inlet, environment temperature, outlet pressure are set as same as Section 3.3.1. The selection of position plane takes every 30 mm along the tube, a total of 7 planes. Figure 9a illustrates the pressure distribution at different planes in the fluid domain model. It can be found from the pressure nephogram that the pressure in the PCMT is symmetrically distributed and gradually decreases from the surrounding to the center. At the same time, the surface pressure is almost constant along the radial direction of the pipe. Figure 9b shows the trajectory of water flow through PCMT. It can be found that the water filters from the outer surface of the PCMT, flows through the ceramic membrane in the inner cavity to the outlet. The flow velocity reaches the maximum at the outlet center. Figure 9c,d show the velocity distribution at different planes in the fluid domain model. The velocity is also symmetrically distributed and decreases from the center to the surrounding. The velocity of water in the ceramic membrane tube is increasing with the decreasing of distance to the outlet, reaches the maximum velocity of 2.301 m/s at the center of outlet.

### 3.4. Experimental Verification of Water Filtering

In order to verify the filtration effect of proposed ceramic membrane tube, a water filtration performed as shown in Figure 10. The membrane tube sample was equipped in the filter device, and the water flowed through the filter and to the beaker. The stopwatch set the time limits, and the analytical balance measured the weight water in beaker every 30 s. Four membrane tube samples were tested with the device in Figure 10a, and the experimental results were shown in Figure 10b. The outlet flow increases with the inlet pressure from 0.01 MPa to 0.1 MPa. The water flow rage reaches its saturation when the inlet pressure larger than 0.1MPa. As the porosity of ceramic membrane tube increases from 31.33% to 48.31%, the outlet flow increases from 0.25 L/min to 0.56 L/min, when the inlet pressure reaches to 0.1 MPa.

Experimental results indicate that the filtration effect of porous ceramic membrane tube increases with the increases of porosity at constant feed mass flow. The accuracy of simulation results has been verified by experiment, which is important for the future preparation of porous ceramic membrane tubes used for filtration.

## 4. Conclusions

In this work, the single-channel Al_2_O_3_-based PCMT was prepared using different grain sizes of Al_2_O_3_ powders by extrusion molding process combined with the traditional solid-phase sintering method. The porosity and pore size increase with the increase in raw particle size but decrease with the increase in sintering temperature. The simulation analysis on the porosity on the water filtration was also presented. It can be concluded that when feed mass flow is constant, the transmembrane pressure decreases with the increase in porosity, while the velocity change at the outlet was not obvious. The water filtration effect is better with the porosity being 39.64%, the transmembrane pressure reaches to 77.83 kPa and maximum velocity is 2.301 m/s. The pressure in each section is almost constant, and the velocity of water increases with the decreasing of distance to the outlet. The porosity of PCMT was regulated by extrusion molding process, changing grain size of raw particles and sintering temperature. The water filtration was analyzed by Fluent simulation and experiments, which indicate the single-channel Al_2_O_3_-based PCMT has a great potential for water treatment field.

## Figures and Tables

**Figure 1 membranes-12-00390-f001:**
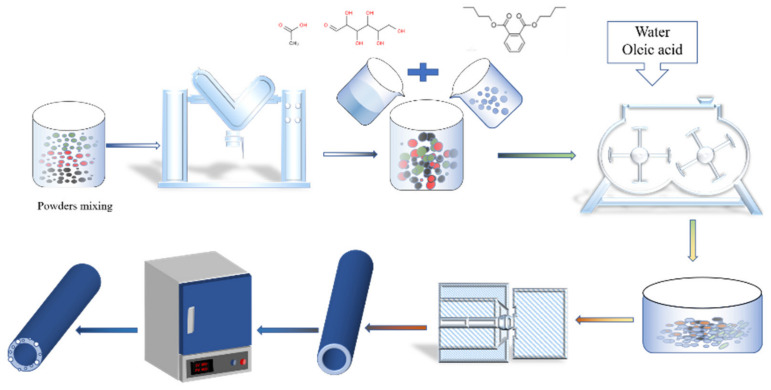
Schematic diagram illustrating the preparation process of single-channel Al_2_O_3_-based PCMT.

**Figure 2 membranes-12-00390-f002:**
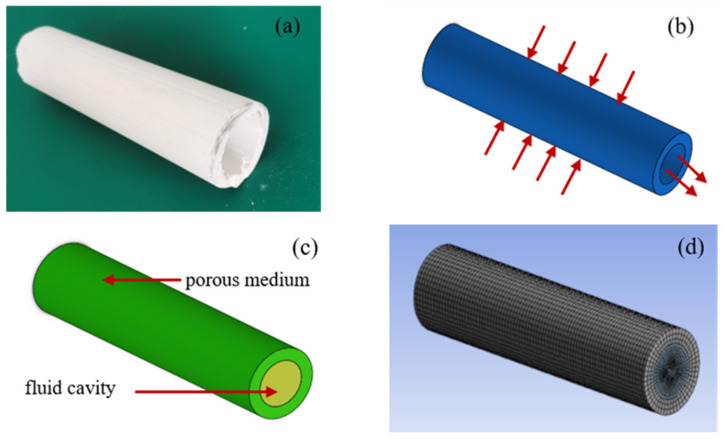
Sample and model of PCMT, (**a**) sample; (**b**) geometric model; (**c**) fluid domain model; (**d**) meshing.

**Figure 3 membranes-12-00390-f003:**
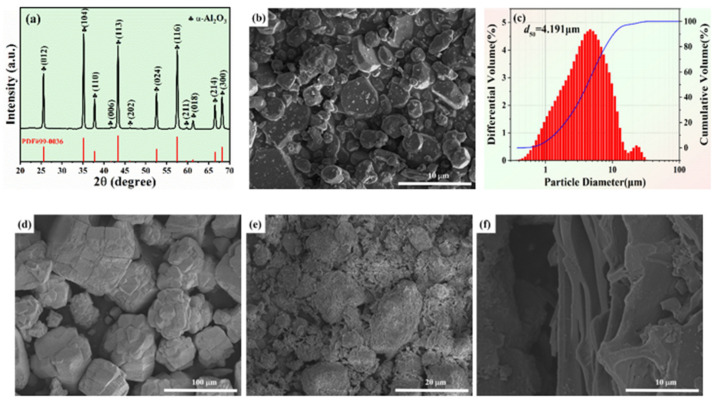
(**a**) The XRD patterns; (**b**) the SEM micrograph; (**c**) the particle diameter of Al_2_O_3_ powders with grit designation of 1500#; (**d**–**f**) the SEM micrograph of raw particles: (**d**) Al_2_O_3_ powders with grit designation of 80–150#, (**e**) kaolin particles, and (**f**) carbon powders.

**Figure 4 membranes-12-00390-f004:**
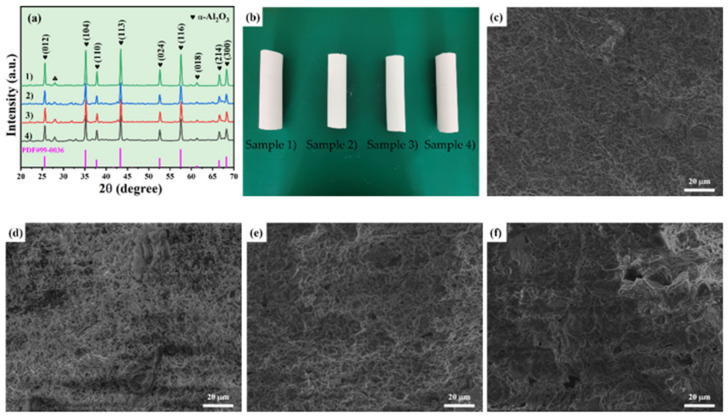
(**a**) The XRD patterns of Al_2_O_3_-based PCMT sintered at 1150 °C using alumina powders with different grit designations: (1) 1500#, (2) 300~325#, (3) 150~250#, and (4) 80~150#; (**b**) the digital photography of single-channel porous alumina ceramic membrane tubes (sample 1–4); (**c**–**f**) the surface SEM images of porous alumina ceramic membrane tubes with different grit designations at 1200 °C: (**c**) 1500#; Sample 1; (**d**) 300~325#, Sample 2; (**e**) 150~250#, Sample 3; and **(f**) 80~150#, Sample 4.

**Figure 5 membranes-12-00390-f005:**
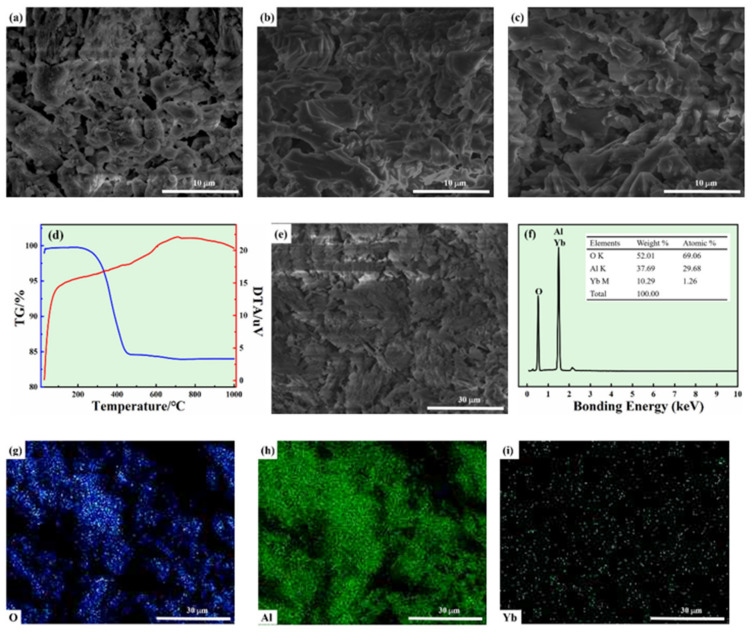
(**a**–**c**) The surface SEM images of porous Al_2_O_3_ ceramic membrane tubes at different temperatures; (**d**) thermogravimetric and differential analyses of green sample; (**d**–**i**) elemental distribution of porous Al_2_O_3_ ceramic membrane tubes prepared at different temperatures: (**a**) 1100 °C; (**b**) 1150 °C; (**c**) 1200 °C; and (**e**–**i**) 1200 °C.

**Figure 6 membranes-12-00390-f006:**
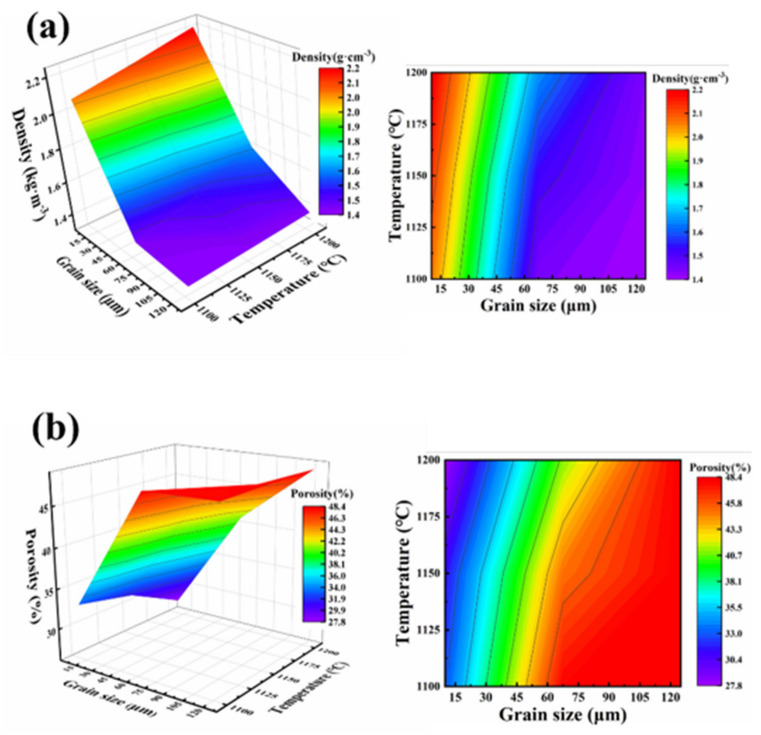
(**a**) Density distribution of porous Al_2_O_3_ ceramic membrane tubes with different grain sizes and sintering temperatures; (**b**) Porosity distribution of porous Al_2_O_3_ ceramic membrane tubes with different grain sizes and sintering temperatures.

**Figure 7 membranes-12-00390-f007:**
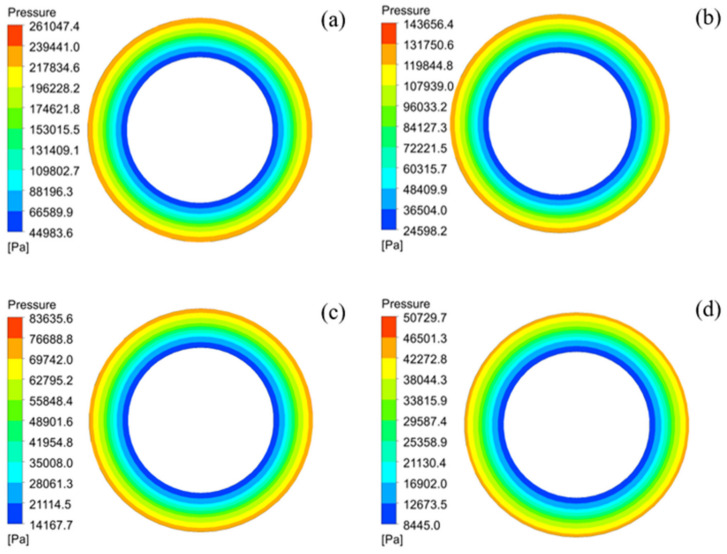
Distribution of flow pressure on the surface of PCMT with different porosity: (**a**) 30%, (**b**) 35%, (**c**) 40% and (**d**) 45%.

**Figure 8 membranes-12-00390-f008:**
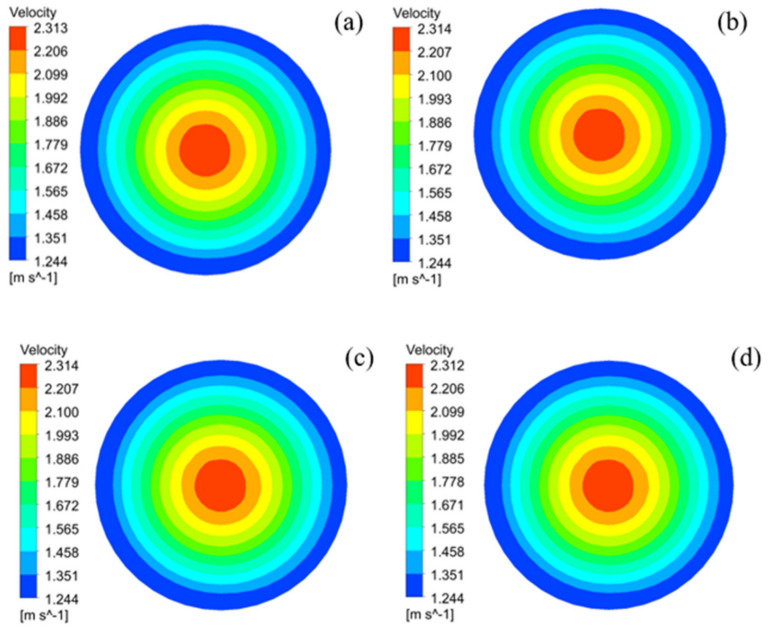
Distribution of flow velocity on the outlet surface of PCMT with different porosity: (**a**) 30%, (**b**) 35%, (**c**) 40% and (**d**) 45%.

**Figure 9 membranes-12-00390-f009:**
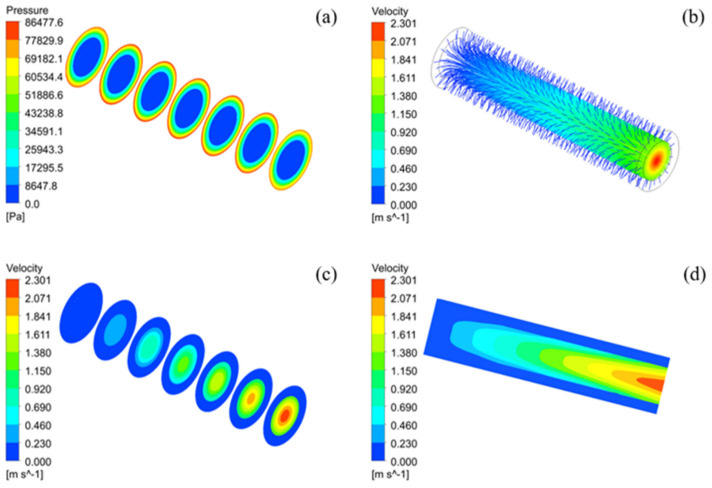
Pressure, flow field and velocity distribution in the fluid domain model: (**a**) pressure distribution, (**b**) flow field, and (**c**,**d**) velocity distribution.

**Figure 10 membranes-12-00390-f010:**
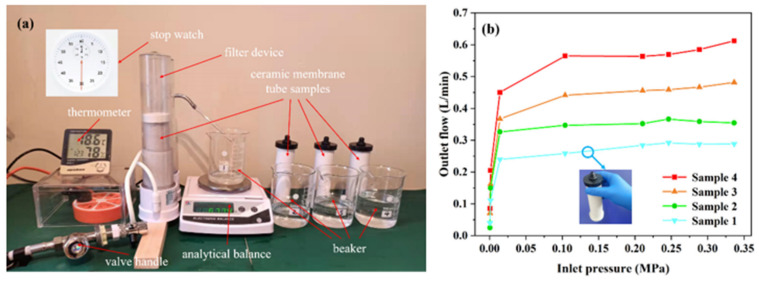
(**a**) Construction of water filtration test system; (**b**) the results of outlet flow for different samples at different inlet pressure.

## Data Availability

Not applicable.

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
