# Peer review of "Fabrication, Characterization and Drainage Capacity of Single-Channel Porous Alumina Ceramic Membrane Tube"

_membranes, 2022, doi:10.3390/membranes12040390_

Round 1

Reviewer 1 Report

The document describes the preparation and characterization of alumina ceramic membranes to be used in water treatment. The effect of the material grain size on the final product is the main variable tested. 

Few comments for the document:

Please consider relocating figure 2 to be placed in the results section. 

The description of figure 4 is confusing since it is difficult to identify when it is related to the subpanel a) or the rest of the figure. 

In figure 5, there are two subpanels identified as f) please correct. 

Although the simulation can give useful information on the performance of the membranes, validation or verification of the results with experimental  results would be useful

Reviewer 2 Report

This manuscript has great potential in my opinion. There are several aspects that the authors can work on. 

  1. The authors like to write long sentences. However, long sentences are not necessary in scientific writing. Please proofread the manuscript and break down long sentences for better delivering information.
  2. The introduction section did not cite enough journal articles, and the current articles are just listed and summarized. There is not a logic backbone in the Introduction. Please add more reference and consider organizing the paragraphs. 
  3. The formulation came from no where. No explanation why choose this specific formulation. 
  4. The Celsius unit is not consistent with the format of the rest of the draft.
  5. Figure 2 was discussed on line 162, but at this point, figure 3 and figure 4 have been discussed. The sequence is confusing. 
  6. I enjoy the simulation work of porosity, density and water filtering of the membranes. May the authors add experimental parts of this? With experimental data, the simulation work will be even more meaningful. 

Round 2

Reviewer 1 Report

No further suggestions

Reviewer 2 Report

The authors addressed all my comments. The experimental section added is okay but I hope the authors can improve the quality of the hand-on experiments in the future while keep the good word of simulation.